# Physician estimates of the feasibility of preserving the dying for future revival

Ariel Zeleznikow-Johnston[1]*, Emil F. Kendziorra[2], Andrew T. McKenzie[3]*

**1** School of Psychological Sciences, Monash University, Melbourne, Australia, **2** European Biostasis Foundation, Riehen, Canton of Basel-Stadt, Switzerland, **3** Apex Neuroscience, Salem, Oregon, United States of America

* arielzj.phd@gmail.com (AZ-J); amckenzie@apexneuro.org (AM)

## Abstract

### Objectives

To assess US physicians' estimates of the feasibility of preservation procedures (cryopreservation, aldehyde-based fixation) for enabling future revival, their views on clinical interventions to improve preservation outcomes, and their positions on ethical and legal frameworks surrounding these end-of-life options.

### Design

Cross-sectional survey conducted October 2025.

### Setting

Sermo, an online platform for verified US healthcare professionals.

### Participants

334 physicians comprising 150 primary care physicians (general practice, internal medicine, family medicine) and 184 other specialists.

### Main outcomes and measures

Estimated probability that preservation retains neurally-encoded information sufficient for future revival; support for pre-mortem anticoagulation and pre-cardiac arrest procedure initiation; views on compatibility with patient-centered care.

### Results

Among 334 physicians, the median estimated probability that preservation under ideal conditions could retain sufficient neural information for future revival was 25.5%. Overall, 27.9% found preservation somewhat or very plausible for enabling future revival; 47.0% found it somewhat or very implausible. Most physicians (70.7%) supported prescribing anticoagulants to terminal patients to improve preservation quality;

provided the original author and source are credited.

**Data availability statement:** A list of the survey questions is available here: https://osf.io/awgk7/files/jn27a The full set of participant response data is available here: https://osf.io/awgk7/files/hrm76 The coded participant responses are available here: https://osf.io/awgk7/files/zt9cs.

**Funding:** Research funding for participant payments in this study was supported by a CryoDAO grant (2025.1). https://www.cryodao.org/ One of the authors, Emil Kendziorra, is a board member of CryoDAO. CryoDAO provided support only in the form of participant payments, but did not have any additional role in the study design, data collection and analysis, decision to publish, or preparation of the manuscript. The specific roles of these authors are articulated in the 'author contributions' section.

**Competing interests:** I have read the journal's policy and the authors of this manuscript have the following competing interests: Andrew McKenzie is an employee of Sparks Brain Preservation, a non-profit brain preservation organization, and a director of Apex Neuroscience, a non-profit research organization. Emil Kendziorra is a shareholder and CEO of Tomorrow Bio, a biostasis provider, President of the Board of the European Biostasis Foundation, a non-profit research foundation, a shareholder and director at Oxford Cryotechnology, Inc., a cryopreservation research organization, and a board member at CryoDAO, a Swiss research association.

11.7% opposed. For patients choosing preservation in combination with medical assistance in dying, 44.3% supported initiating preservation prior to cardiac arrest; 28.8% opposed. Most (58.1%) agreed preservation could be consistent with compassionate care (20.1% disagreed), and 49.1% reported comfort with patients choosing preservation (30.0% uncomfortable). Familiarity with preservation correlated with higher probability estimates ($\rho = 0.26$; $p < 10^{-3}$), while end-of-life discussion frequency correlated with support for pre-cardiac arrest procedures ($\rho = 0.18$; $p = 0.003$).

## Conclusions

US physicians assigned a median 25.5% probability to preservation retaining neural information under ideal conditions in a manner potentially compatible with future patient revival. The majority support for pre-mortem anticoagulation and plurality support for pre-cardiac arrest initiation indicate that many physicians would consider accommodating patient requests for preservation-enhancing interventions. These findings may inform development of clinical guidelines, though the speculative nature of the estimates warrants consideration.

## Introduction

Medicine often faces a tragic temporal mismatch: while approximately 70% of terminally ill patients have a strong will-to-live despite their impending deaths [1], treatments that could meaningfully extend their lives may remain years or decades away from development. This gap between current therapeutic limitations and future medical possibilities has prompted some patients to consider 'preservation procedures': interventions designed to preserve their brains and other biological structures after legal death in the hope of future revival when effective treatments become available [2].

Preservation procedures, also known as 'biostasis' or 'cryonics', currently employ two primary approaches: low temperatures combined with cryoprotective agents with the goal to cryopreserve tissue and minimize ice crystal formation [3], and aldehyde-based fixation, which crosslinks proteins to prevent molecular decay [4]. Both methods aim to preserve the neural architecture encoding memory and identity, based on the neuroscientific understanding that personal identity depends primarily on structural brain preservation [2,5]. While no preserved patient has yet been revived, its medical plausibility draws support from procedures like deep hypothermic circulatory arrest, where patients routinely recover full neurological function after 30 minutes of cardiac arrest and electrocerebral silence [6]. Current preservation organizations report several hundred patients preserved globally, with thousands more signed up for future preservation [7].

The clinical community lacks consensus on these procedures, creating challenges for physicians whose patients inquire about preservation as an end-of-life option. In particular, evaluating the success probability of preservation procedures presents a

unique clinical challenge: definitive validation requires successful revival, which if possible at all may be decades away. Without the possibility of near-term experimental validation, assessment must rely on expert judgment about whether current preservation methods adequately maintain neural information. In a recent survey of neuroscientists, the median respondent assigned current preservation methods, performed under ideal conditions, a 40% probability of preserving an individual's brain adequately to be compatible with at least one potential revival method [8].

This current study presents the first systematic survey of hundreds of US physicians' assessments of preservation feasibility, their views on clinical interventions that could improve outcomes, and their positions on the ethical and legal frameworks needed to guide practice.

## Methods

The survey was distributed in October 2025 to physicians registered in the United States of America via Sermo, an online community for verified healthcare professionals. The survey and its implementation were reviewed by the Pearl Institutional Review Board and received an exemption determination (#2025−0579).

### Survey questions

The survey consisted of 5 initial demographic questions and 25 main questions. Most questions were mandatory for completion, except those that asked participants to optionally provide additional commentary on their responses. The main section was preceded by a page of information providing contextual information and definitions required for the questions that followed.

Survey items were developed collaboratively by the study authors to address three domains: (1) the perceived feasibility of preservation procedures, (2) clinical interventions that could improve preservation outcomes, and (3) the ethical and legal standing of preservation as an end-of-life option. The final set of items was selected to balance comprehensive coverage of these domains against survey length constraints. Prior to deployment, the survey underwent informal face validity testing with five physician colleagues. Based on their feedback, questions were iteratively revised to improve clarity and comprehension.

The full text of the survey can be found here: https://osf.io/awgk7/files/jn27a.

### Recruitment and fielding

Participants were recruited through Sermo, with physicians on the platform having had their medical registration verified by the platform administrators [9].

We aimed to recruit a cohort of 330 physicians, consisting of 150 with primary care specialties and 180 in other specialties. Primary care physicians included only physicians who self-identified as specialising in general practice, internal medicine, or family medicine, with all other physicians being classed as other specialists. Additionally, we set a maximum for the number of other specialists we would recruit from different specialties: neurology (40), neurosurgery (40), psychiatry (40), intensive care (30), anesthesiology (30), emergency medicine (30), palliative care (30), pathology (20), radiology (20).

Participants on Sermo are invited to participate in surveys in several ways: some opt-in to email invites to surveys, while others view surveys that are available when they log into their homepage. Our invitation contained the survey's title ('Understanding your thoughts on preservation procedures post declaration of death'), length (around 15 minutes), and payment ($28 for primary care physicians, $43 for other specialists). On accepting the invitation, participants provided written electronic consent to participate. On completion, payments were credited to their Sermo account, which could be withdrawn via check or cashcard at any time.

Participant responses were rejected from the survey for any of the following: 1) if they submitted the survey in under 3 minutes from commencement (the minimum time required to comprehend the questions based on pre-deployment

validation); or 2) if they selected either Michigan or Vermont as their state of practice (as payments for medical surveys are illegal in those states). Participant responses by US state are provided in Supplementary Figure 1 in S1 File.

This survey was initially posted to the platform on 2025-10-21 but initial completion was limited to 34 participants. As no issues were observed with initial deployment, the survey was opened to a full cohort on 2025-10-22, which was completed on 2025-10-24.

598 individuals initially responded to the invitation. Of these, 2 (0.3%) failed to complete the initial demographic questions, 6 (1%) were screened out based on state of practice, and 208 (35%) were stopped after the initial demographic questions because their specialty had met quota. Of the 382 who commenced the main survey, 7 (2%) failed to complete, 41 (11%) were excluded for attempting to submit in under 3 minutes, and 334 (87%) completed successfully.

## Analysis

Data analysis was performed using R statistical software, version 4.3.1. Statistical analyses included Spearman correlation analyses. All reported p-values are two-tailed.

To assess the relationships between participants' responses to different questions, we performed a rank correlation analysis across a subset of the relevant variables in our dataset. These included demographic variables (age, gender) and main survey question responses. In order to include them in the analysis, ordinal variables were coded numerically. For example, gender was coded as 1 (male) or 0 (female), while end-of-life discussion frequency was coded from 1–5 ('never' to 'very frequently (weekly or more)'). After calculating correlation coefficients and corresponding p-values for all variable pairs, we applied a Benjamini-Hochberg correction to control for multiple comparisons with a false discovery rate of 0.05. Only the correlations that remained significant after this correction are shown in Fig 4.

Open-ended responses were extracted from the optional free-text comment fields and filtered to exclude blanks, "None" entries, and responses fewer than 10 characters. The complete response corpus was reviewed in full by a large language model (Claude Sonnet 4.6, Anthropic, March 2026), which identified thematic categories and cross-cutting patterns. To enable reproducible per-comment coding, the language-model-identified themes were operationalised as keyword-based regular expression rules across eight categories (scientific skepticism, patient autonomy, religious/spiritual/framing, equity and access concerns, ethical/moral concerns, legal/regulatory concerns, supportive/optimistic, and death acceptance framing), with a single response able to receive multiple labels. Additional emergent themes not amenable to simple keyword rules were operationalised as more complex pattern sets and are documented in the supplementary coded dataset (open_ended_llm_coded.csv). All qualitative findings are anchored to specific respondent IDs and question labels in that file to permit independent verification.

## Results

### Preservation's estimated probability of success

Participants were asked about the probability of success of a preservation procedure in two different ways.

First, participants were asked "How plausible do you find the idea that preservation could potentially allow for some form of revival in the future?" 27.9% of respondents answered 'somewhat plausible' or 'very plausible', while 47% responded 'somewhat implausible' or 'very implausible' (Fig 1a).

Second, participants were asked to imagine a scenario where a patient is preserved within minutes of cardiac arrest, with follow-up imaging and brain biopsies showing intact brain structures down to the synaptic level. They were then asked to assign a probability that "a significant amount of the neurally-encoded information required for long-term memory and personality is still preserved in their brain, such that it may be technologically possible to revive this patient, even in the distant future?" The median physician response was 25.5% (Fig 1b), which was roughly consistent across specialties (Fig 1c).

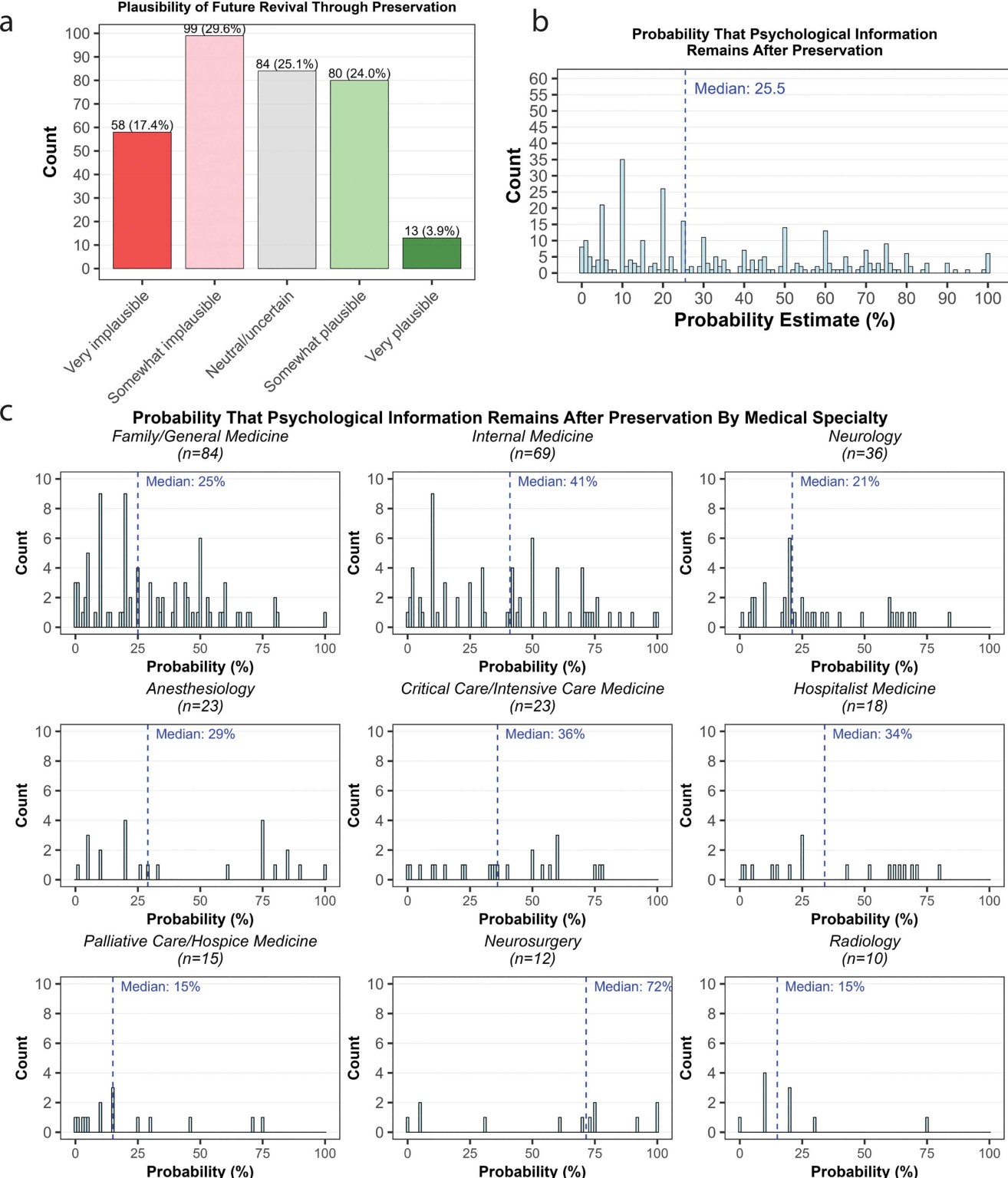

**Fig 1. Plausibility and probability estimates for whether preservation may enable future revival. a)** Responses to the question 'How plausible do you find the idea that preservation could potentially allow for some form of revival in the future?' **b)** Responses to the question '*During end-of-life planning discussions, an elderly, cognitively-intact patient expressed a desire for preservation. Imagine that later they suffer a cardiac arrest and are*

*successfully preserved within minutes of the event. Follow-up imaging and brain biopsies show intact brain structure down to the synaptic level, including the spatial distribution of key biomolecules.* How probable do you think it is that a significant amount of the neurally-encoded information required for long-term memory and personality is still preserved in their brain, such that it may be technologically possible to revive this patient, even in the distant future?' **c)** Same as (b) but broken down by specialty.

### Interventions to improve preservation's probability of success

The perfusability of a patient's brain and body is considered a necessity for high-quality preservation. This can be hampered by clots that form shortly after cardiac arrest. Anticoagulants such as heparin, administered pre-arrest, could improve the quality of preservation commenced post-arrest. When asked if it should be allowable to prescribe heparin to an imminently terminal patient (i.e., to improve the quality of their preservation rather than treat their current illness), 70.7% of respondents said this probably or definitely should be allowed, while 11.7% said this probably or definitely should not be allowed (Fig 2a).

Currently, due to legal concerns, preservation procedures are only begun after cardiac arrest and declaration of death, even for terminally ill patients who choose preservation in conjunction with medically-assisted dying. Under this legally postmortem condition, 68% of respondents said the procedure should probably or definitely be legal, while 11.4% stated it should probably or definitely not be legal. However, this delay can significantly compromise preservation quality compared to animal studies where preservation can begin under deep anesthesia prior to cardiac arrest. Participants were also

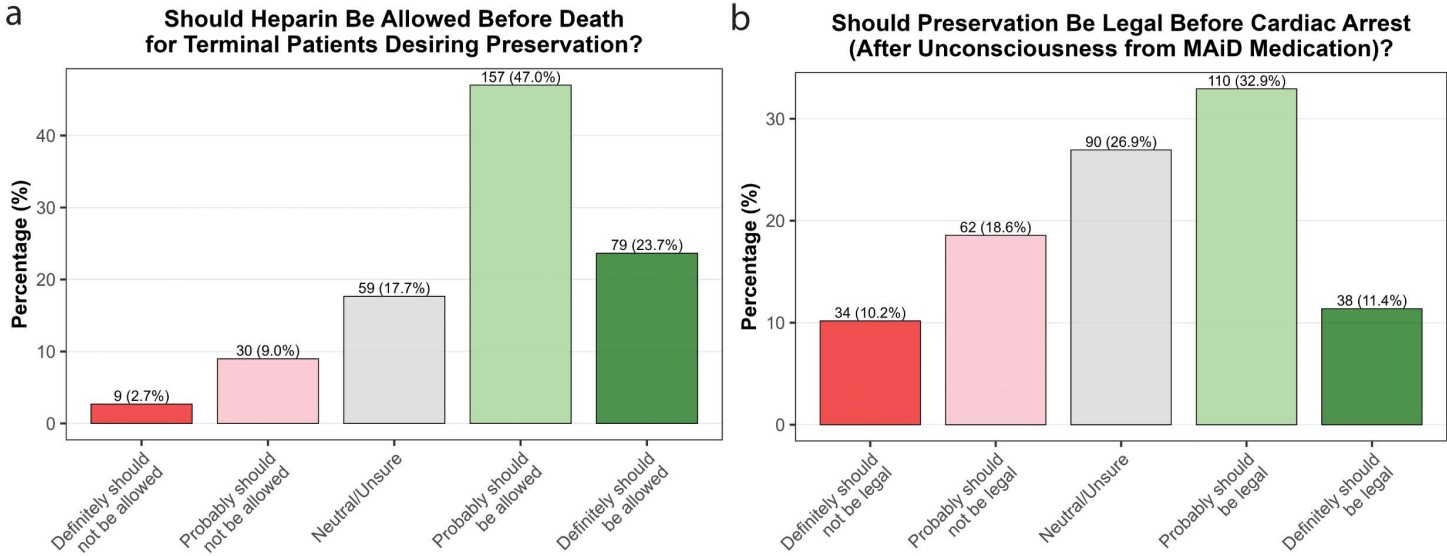

**Fig 2. Views on interventions to improve preservation success. a)** Responses to the question 'Successful preservation requires a patient's brain and body to be perfusable, which can become difficult following cardiac arrest due to clots. Anticoagulants (e.g., heparin) provided before cardiac arrest (i.e., before legal death) have been found to substantially reduce this problem. Do you think it should be allowable to prescribe heparin to an imminently terminal patient who desires this treatment and wishes to be preserved following legal death?' **b)** Responses to the question 'Currently, preservation procedures can only legally begin after cardiac arrest and declaration of death, even for terminally ill patients who choose medically-assisted dying. This delay can significantly compromise preservation quality compared to animal studies where preservation can begin under deep anesthesia prior to cardiac arrest. Consider this scenario: A terminally ill patient who has chosen medically-assisted dying specifically requests preservation and wants the procedure to begin as quickly as possible for optimal preservation quality. Imagine the steps could proceed as follows: 1. Patient self-administers or receives medically-assisted dying medication (which will cause cardiac arrest) 2. Patient becomes unconscious from the medication 3a. Preservation procedures can commence now, while the heart is still beating but death is inevitable and imminent OR 3b. Preservation procedures must wait until cardiac arrest occurs and death is declared. Do you agree it should be legal to begin preservation procedures after the patient is unconscious from medically-assisted dying medication but before cardiac arrest, if the patient has explicitly requested this?'.

presented with a hypothetical scenario where a terminally ill patient requests preservation in conjunction with medically-assisted dying, and asked whether it should be legal to begin preservation procedures after the patient is unconscious but prior to cardiac arrest. 44.3% of respondents said this should probably or definitely be legal, as opposed to 28.8% who stated this should probably or definitely not be legal (Fig 2b).

## Views on the ethics of preservation

Preservation, although potentially relevant to sensitive end-of-life discussions, is a speculative and confronting procedure. As such, we were interested in the degree to which physicians saw the procedure as compatible with compassionate, patient-centered care. 58.1% of respondents somewhat or strongly agreed that it was, while 20.1% somewhat or strongly disagreed (Fig 3a).

As a speculative therapy with an uncertain probability of success, patient requests for preservation arguably resemble requests for experimental drugs, as occurs under the Early Access to Medicines Scheme (EAMS) in the UK, the Compassionate Use programs in Europe, or the Expanded Access programs in the US. In these circumstances, unvalidated treatments are provided to patients with no other non-palliative treatment options. When participants were asked about the ethical standing of preservation in comparison to other experimental treatments, 39.2% said they were similarly ethically problematic, 28.4% stated they had no significant difference, 18.3% said preservation was more ethically problematic, 8.1% said preservation was less ethically problematic, and 6.0% said the two were incomparable (Fig 3b).

Physician decisions to provide or prescribe treatments could be influenced by their own comfort with patients choosing such treatments. We found that 49.1% of doctors stated they were somewhat or very comfortable with patients choosing preservation, while 30% said they were somewhat or very uncomfortable (Fig 3c).

## Influences of familiarity with preservation & frequency of end-of-life discussions

Analysis of the correlations between survey responses demonstrated internal consistency in participant response to questions, as evidenced by the positive correlation between reported plausibility of preservation enabling revival and the probability that preservation retains critical psychological information ($\rho = 0.45$, $p < 10^{-3}$), as well as the correlation between whether physicians view a preservation procedure as compatible with compassionate care and their personal comfort with patients making use of the procedure ($\rho = 0.55$, $p < 10^{-3}$) (Fig 4).

More interestingly, this analysis also revealed several significant relationships between physicians' clinical practice and their views on preservation. Notably, the frequency with which they engaged in end-of-life discussions was positively correlated with whether they thought it should be legal to commence preservation procedures prior to cardiac arrest ($\rho = 0.18$, $p = 0.003$), the degree to which they were comfortable with a patient requesting the procedure ($\rho = 0.16$, $p = 0.007$), whether they believed such a patient had an acceptable attitude towards death ($\rho = 0.13$, $p = 0.030$), and also the physician's self-assessed familiarity with preservation procedures ($\rho = 0.26$, $p < 10^{-3}$). Additionally, physician self-assessed familiarity with preservation procedures was also positively correlated with both their confidence that the procedure might succeed (plausibility: $\rho = 0.23$, $p < 10^{-3}$, probability of retaining critical information: $\rho = 0.26$, $p < 10^{-3}$), how ethically acceptable they thought preservation was (personal comfort: $\rho = 0.24$, $p < 10^{-3}$; ethical standing vs other experimental treatments: $\rho = 0.13$, $p = 0.032$), and whether they thought it should be legal to commence the procedure prior to cardiac arrest ($\rho = 0.16$, $p = 0.007$).

Physician gender and age were also associated with some attitudinal differences. Male physicians were more familiar with preservation procedures ($\rho = 0.21$, $p < 10^{-3}$), more personally comfortable with patients requesting them ($\rho = 0.17$, $p = 0.006$), more frequently engaged in end-of-life discussions ($\rho = 0.19$, $p = 0.002$), and more supportive of heparin administration prior to cardiac arrest ($\rho = 0.13$, $p = 0.045$). Older physicians similarly reported greater familiarity with preservation procedures ($\rho = 0.23$, $p < 10^{-3}$) and greater support for heparin administration ($\rho = 0.13$, $p = 0.038$). No other significant age or gender based attitudes were observed.

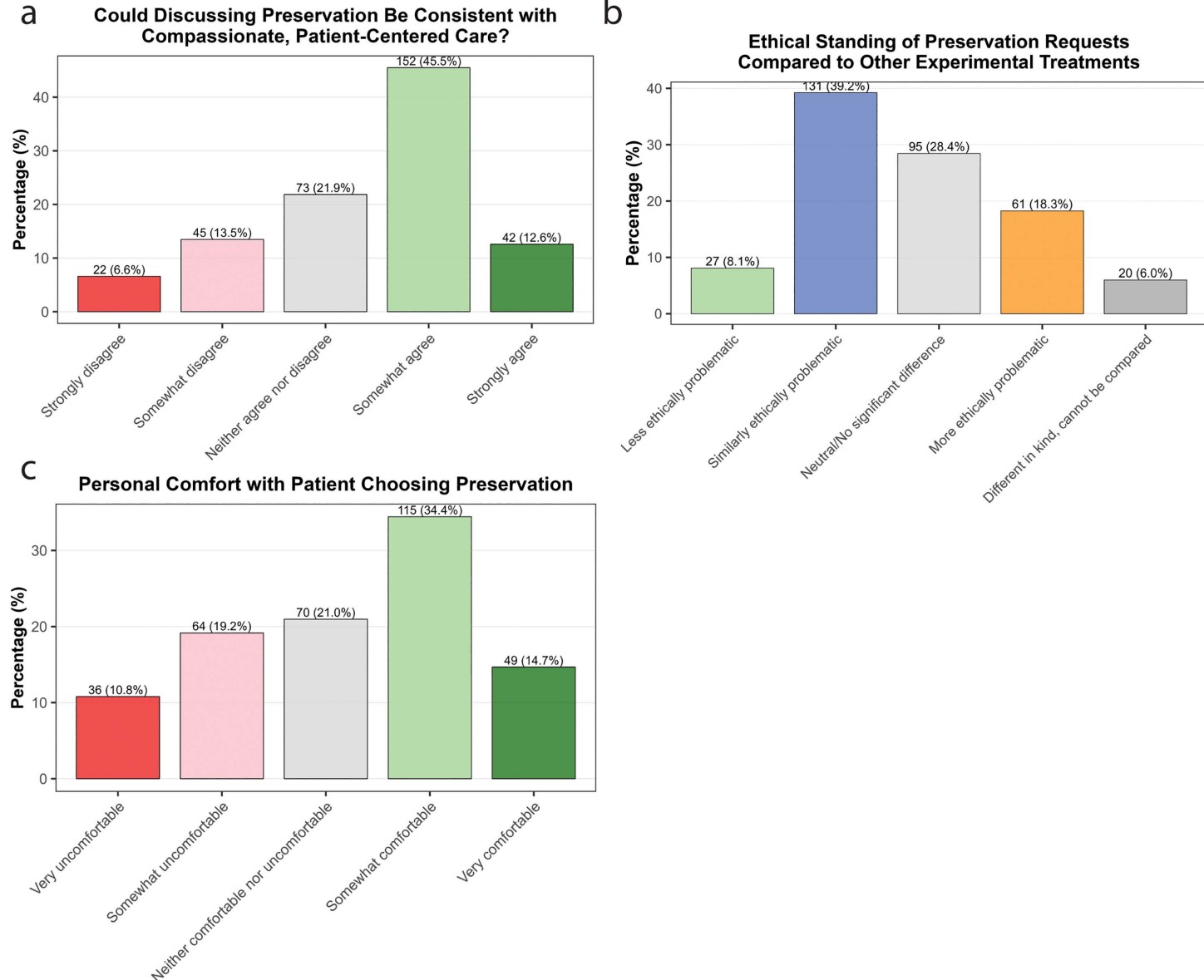

**Fig 3. Views on the ethics of preservation. a)** Responses to the question 'Consider a terminally ill patient expressing a strong will-to-live as well as strong concerns about death, specifically about "ceasing to exist forever." They report that the possibility of preservation, even with uncertain odds of revival, significantly reduces their distress and improves their quality of life in their remaining time. In such cases, do you believe discussing preservation as an option with them and helping them to access it could be consistent with providing compassionate, patient-centered care?' **b)** Responses to the question 'Compared to other experimental medical treatments with uncertain outcomes (e.g., Expanded Access/Compassionate Use, Right to Try), how do you view the ethical standing of patient requests for preservation?' **c)** Responses to the question 'Setting aside professional and legal considerations, how do you personally feel about the choice to pursue preservation, as made by a cognitively-intact, terminally-ill patient paying for the procedure out-of-pocket?'.

## Additional commentary through open-ended responses

Of 334 respondents, 221 (66.2%) provided at least one open-ended comment, contributing 1,487 comment instances across 12 questions (mean 6.7 per commenting respondent). The most frequently coded keyword themes were supportive/optimistic (9.8% of responses), patient autonomy (11.4%), legal/regulatory concerns (10.6%), ethical/moral concerns

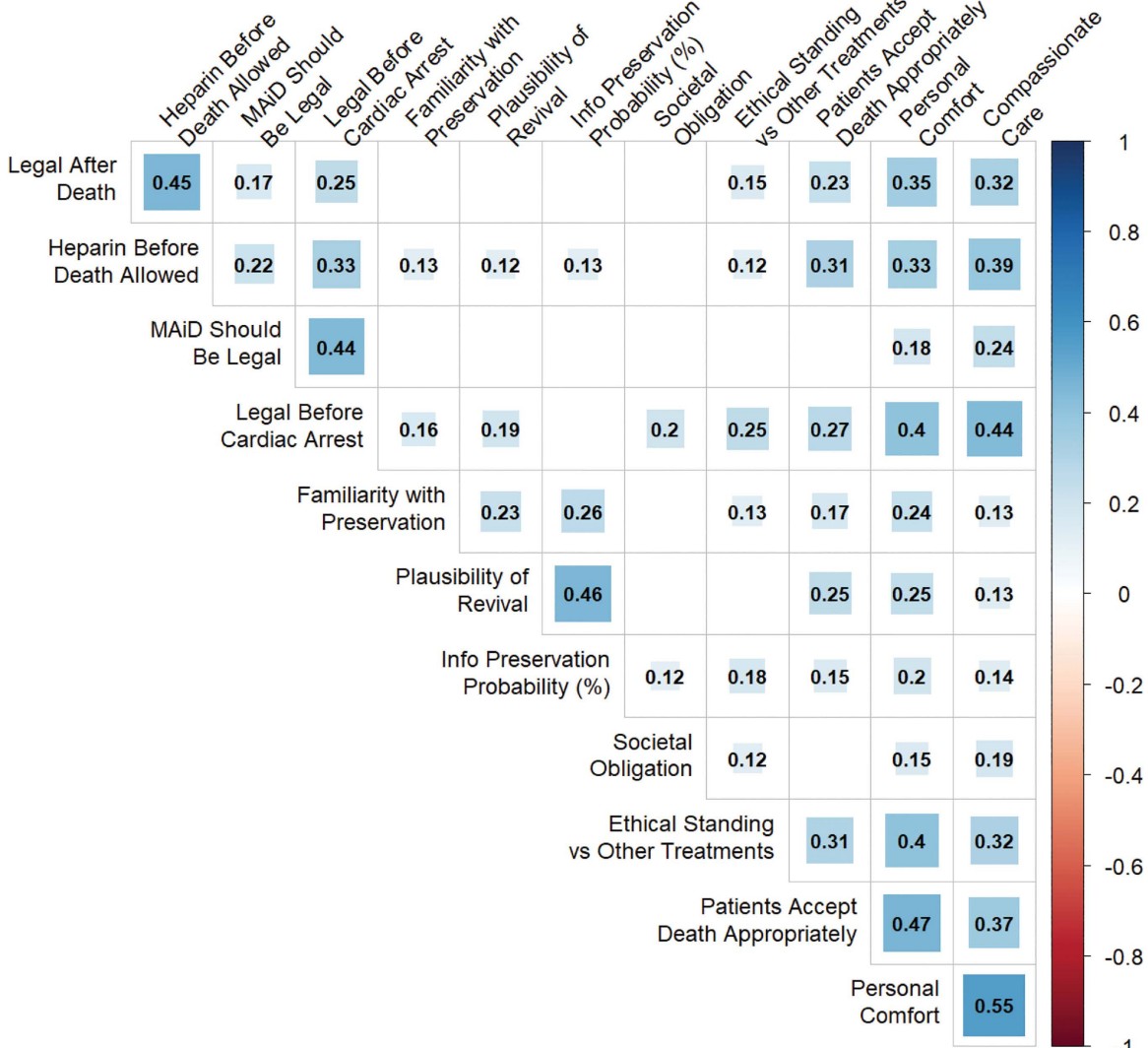

**Fig 4. Correlations between participant responses.** *EOL Discussion Frequency* corresponds to how frequently they engage in end-of-life care discussions. *Legal After Death* corresponds to their beliefs on whether it should be legal to perform preservation procedures after a patient is declared dead. *Societal Obligation* corresponds to the degree to which they believe society has a moral or ethical obligation to provide preservation services to all who desire them, should revival prove highly probable. *Patients Accept Death Appropriately* corresponds to the extent to which they disagree with the statement 'Patients who request preservation are not accepting death appropriately.'.

(10.0%), and equity and access concerns (6.9%). Full-corpus qualitative review identified three cross-cutting tensions not well captured by keyword coding. First, an autonomy–equity tension: 22 of 221 commenting respondents (10%) raised both patient-autonomy arguments and wealth-access concerns across their comments, reflecting a conflict internal to the autonomy position itself: "If universally acceptable, preservation should be equally accessible to people regardless of their socioeconomic status; otherwise, this will cause significantly negative impact to human being society and history." (respondent 188, Q25). Second, a false-hope vs. compassionate-hope debate (present in 19 comments, 1.3%), most prominent in responses to Q20. Third, a consciousness and personal identity problem (present in 23 comments, 1.5%), who questioned whether revival would restore a continuous personal identity regardless of structural preservation

quality: "We don't understand how consciousness works. It's very difficult to restore something we don't understand" (respondent 303, Q4).

## Discussion

This study provides the first formal survey of the medical community's views on the practice of preserving dying patients for future revival. Physicians reported a median probability estimate of 25.5% that preservation could retain neural information in a manner compatible with future revival, with roughly a quarter finding it plausible and half finding it implausible. A substantial majority supported pre-mortem anticoagulation to improve preservation quality, and around half supported initiating preservation prior to cardiac arrest when combined with medically-assisted dying. Most physicians viewed preservation as compatible with compassionate care and ethically comparable to other experimental treatment access schemes. Correlation analysis revealed that the more frequently physicians engaged in end-of-life discussions the more comfortable they were with the procedure, and that the more familiar they were with preservation procedures the more likely they thought they were to work.

The median physician estimate of 25.5% for preservation success, though speculative and without empirical backing, naturally prompts comparisons to accepted medical procedures with similar or lower success rates. For example, 22.0% of patients receiving cardiopulmonary resuscitation for out-of-hospital cardiac arrest survive to admission [10], emergency department resuscitative thoracotomy has a survival-to-discharge rate of 7.8% [11], and salvage therapy for refractory or relapsed acute myeloid leukemia has a one-year survival rate of under 26% [12] (though note that the epistemic uncertainty of preservation's success is admittedly different to the purely aleatoric uncertainty in cardiopulmonary resuscitation outcomes). It also prompts comparison to other unproven last chance medical interventions, such as those available through the Early Access to Medicines Scheme or the Compassionate Use/Expanded Access programs, given the vast majority of doctors agreed that preservation is ethically equivalent to those schemes. Given most agreed the procedure is compatible with compassionate, patient-centered care and should be legal, it is arguable that clarifying the clinical, legal, and ethical frameworks for use of preservation as an end-of-life procedure would benefit both patients inquiring about these procedures and physicians navigating these requests.

While interventions exist that could substantially improve preservation outcomes, their implementation faces significant legal uncertainty. The duration of ischemia between cardiac arrest and commencement of preservation procedures critically determines outcome quality, with prolonged ischemia leading to progressively worse preservation outcomes, due to both perfusion impairment and the breakdown of cellular structures [13,14]. This has been avoidable in animal models of the procedure that commence perfusion-based preservation under deep anesthesia prior to cardiac arrest, resulting in preserved brain ultrastructure integrity still clearly apparent on electron microscopy [4]. However, starting pre-cardiac arrest preservation in humans is, to the best of our knowledge, not legally permitted anywhere in the world, even in cases where patients request preservation alongside medical assistance in dying. Without successful revival to date, performing preservation prior to legal declaration of death may be viewed as an illegal hastening of death – a determination that admittedly hinges on contested definitions of death and what constitutes "irreversible cessation" of circulatory and brain functions [15]. In our survey, 44.3% of physicians supported (versus 28.8% opposed) allowing pre-cardiac arrest procedures when patients have exhausted treatment options and chosen to end their lives, suggesting potential openness within the medical community to policy frameworks that would permit such interventions under carefully defined circumstances.

Even more modest interventions face similar uncertainty. In our survey, 70.7% of physicians supported prescribing anticoagulants to terminal patients specifically to improve preservation quality – potentially even absent therapeutic benefit for their underlying condition and while accepting a small increased bleeding risk. Yet whether prescribing medications solely to optimize post-mortem procedures falls within accepted medical practice has not been formally established. Given the substantial physician support for this practice, proactive legal clarification would be valuable, particularly as ongoing

disputes about the definition of death in the United States, including recent failures to update the legal definitions, suggest that waiting for organic legal evolution may leave patients and physicians in prolonged uncertainty about permissible practice.

Notably, physicians' familiarity with preservation procedures was positively correlated with both their plausibility and probability estimates for how likely it was that preservation would enable future revival. This correlation admits at least two potential interpretations: increased knowledge of the scientific basis and technical details of preservation may lead to greater confidence in its potential, or alternatively, physicians with a pre-existing interest in life extension may be more likely to familiarize themselves with these procedures. Regardless of causality, this finding suggests that educational initiatives about preservation procedures could influence physician attitudes, particularly given that familiarity also correlated with greater comfort discussing these options with patients and support for pre-cardiac arrest interventions.

The substantial degree of variation in familiarity with preservation procedures (see Supplementary Table 1 in S1 File) suggests that many physicians are unaware of the broader context in which this form of potential life extension has been proposed and criticised. While experiments on freezing and reviving fish were described by Robert Boyle as early as 1665, and the basic idea of preserving the bodies of humans for potential future revival was described by Benjamin Franklin by 1773, the modern movement is generally traced to a book published in 1962 by Robert Ettinger [16–18], which has stimulated further elaborations in the decades since [2,19–21]. These proposals have also been considered in bioethics discussions, both in isolation and in relation to other contentious issues such as euthanasia [22–28]. These discussions have not been entirely theoretical, with questions of legal autonomy as it relates to preservation having already been tested in courtrooms [29]. Separately from questions of morality and legality, philosophers have also debated which revival methods (if any) would actually constitute a form of survival for the preserved individual, with these discussions situated in broader questions on the nature of personal identity [30–33]. There has also been a small amount of social science research into the moral intuitions and personality traits that may drive members of the general public to find preservation more or less desirable and/or socially acceptable [34–37]. Notably, the themes that emerged in participants' open-ended responses – including tensions around patient autonomy and equitable access, debates over whether preservation offers false hope or compassionate hope, and questions about whether revival would constitute genuine survival – closely mirror the central tensions in this broader scholarly discourse.

This study has several limitations that should be considered when interpreting our findings. First, our sample was restricted to US physicians, limiting generalizability to international medical communities where attitudes toward death, experimental treatments, and patient autonomy may differ substantially. Second, we did not make a particular effort to perform regional analysis within the US, and given we lack a large number of responses from each US state, we are underpowered to perform geographic analyses that are more than purely descriptive. Third, while we observed consistent median probability estimates across specialties (ranging from approximately 20–30%), our sample size was insufficient to detect potentially meaningful differences between specialty subgroups. Fourth, our thematic analysis of qualitative comments relied on large language model assisted coding rather than traditional qualitative methods such as grounded theory or thematic analysis by independent human raters, and the emergent themes identified should be interpreted accordingly. Fifth, as we did not run multiple variants of the survey with randomised question presentation order or response order, we cannot entirely rule out acquiescence or question ordering effects. However, we believe the relatively low probability estimate of 25.5% revival success, as well as the substantial opposition described on several items, indicates that acquiescence bias did not dominate participant responses.

This first systematic survey of physician attitudes toward preservation procedures reveals that a substantial proportion of the medical community views these interventions as potentially viable, with a median probability estimate of 25.5%. While this is lower than the 40% probability that preservation could retain neural information as suggested by neuroscientists [8], it is still comparable to many accepted emergency medical procedures. The majority of physicians surveyed support both prescribing medications to optimize preservation outcomes and, in appropriate circumstances, a plurality support

initiating procedures prior to cardiac arrest. Given this physician support and the current legal ambiguities surrounding optimal preservation protocols, it may be appropriate to develop clinical guidelines and legal frameworks to ensure that patients seeking preservation can receive interventions that maximize their probability of successful future revival while maintaining appropriate ethical standards.

## Supporting information

**S1 File. Supplementary Material (Sup Table 1, 2, 3; Sup Fig 1).**
(DOCX)

## Author contributions

**Conceptualization:** Ariel Zeleznikow-Johnston, Emil F. Kendziorra, Andrew T. McKenzie.

**Data curation:** Ariel Zeleznikow-Johnston.

**Formal analysis:** Ariel Zeleznikow-Johnston.

**Funding acquisition:** Emil F. Kendziorra.

**Investigation:** Ariel Zeleznikow-Johnston, Andrew T. McKenzie.

**Methodology:** Ariel Zeleznikow-Johnston, Andrew T. McKenzie.

**Project administration:** Ariel Zeleznikow-Johnston.

**Resources:** Ariel Zeleznikow-Johnston.

**Software:** Ariel Zeleznikow-Johnston.

**Validation:** Ariel Zeleznikow-Johnston.

**Visualization:** Ariel Zeleznikow-Johnston.

**Writing – original draft:** Ariel Zeleznikow-Johnston, Andrew T. McKenzie.

**Writing – review & editing:** Ariel Zeleznikow-Johnston, Emil F. Kendziorra, Andrew T. McKenzie.

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
