## [Decision Letter · Decision Letter 0]

3 Mar 2026

Dear Dr. Zeleznikow-Johnston,

Thank you for submitting your manuscript to PLOS ONE. After careful consideration, we feel that it has merit but does not yet fully meet PLOS ONE’s publication criteria as it currently stands. Therefore, we invite you to submit a revised version of the manuscript that addresses the points raised during the review process.

The reviewers were broadly supportive, and all recommend publication, recognising the study as a timely, rigorous, and an original contribution that addresses an important gap in understanding attitudes towards human preservation. The reviewers recommended shortening and refining parts of the discussion; clarifying terminology (avoiding “cryopreservation” where preservation is not established); contextualising the findings more explicitly within the broader bioethical and legal literature; providing additional detail on the survey design and population; and incorporating at least a brief synthesis of the open-ended responses.

We look forward to receiving your revised manuscript.

Kind regards,

Barry L. Bentley, Ph.D.

Academic Editor

PLOS One

Journal Requirements:

“Research funding for this study was supported by a CryoDAO grant (2025.1).”

“Research funding for participant payments in this study was supported by a CryoDAO grant (2025.1). https://www.cryodao.org/ One of the authors, Emil Kendziorra, is a board member of CryoDAO. CryoDAO played no role in the study design, data collection and analysis, decision to publish, or preparation of the manuscript.”

“Research funding for participant payments in this study was supported by a CryoDAO grant (2025.1). https://www.cryodao.org/ One of the authors, Emil Kendziorra, is a board member of CryoDAO. CryoDAO played no role in the study design, data collection and analysis, decision to publish, or preparation of the manuscript.”

We note that one or more of the authors is affiliated with the funding organization, indicating the funder may have had some role in the design, data collection, analysis or preparation of your manuscript for publication; in other words, the funder played an indirect role through the participation of the co-authors. If the funding organization did not play a role in the study design, data collection and analysis, decision to publish, or preparation of the manuscript and only provided financial support in the form of authors' salaries and/or research materials, please do the following:

1. Review your statements relating to the author contributions, and ensure you have specifically and accurately indicated the role(s) that these authors had in your study. These amendments should be made in the online form.

2. Confirm in your cover letter that you agree with the following statement, and we will change the online submission form on your behalf:

“The funder provided support in the form of salaries for authors [insert relevant initials], but did not have any additional role in the study design, data collection and analysis, decision to publish, or preparation of the manuscript. The specific roles of these authors are articulated in the ‘author contributions’ section.

Reviewers' comments:

Reviewer's Responses to Questions

**Comments to the Author**

1. Is the manuscript technically sound, and do the data support the conclusions?

Reviewer #1: Yes

Reviewer #2: Yes

Reviewer #3: Yes

2. Has the statistical analysis been performed appropriately and rigorously?

Reviewer #1: Yes

Reviewer #2: Yes

Reviewer #3: Yes

3. Have the authors made all data underlying the findings in their manuscript fully available?

Reviewer #1: Yes

Reviewer #2: Yes

Reviewer #3: Yes

4. Is the manuscript presented in an intelligible fashion and written in standard English?

Reviewer #1: Yes

Reviewer #2: Yes

Reviewer #3: Yes

Reviewer #1: This article addresses questions of growing interest to the medical community. As various forms of preservation become more common, physicians and other members of the medical community will increasingly need to respond appropriately to patients who have a personal interest in the subject. As a consequence, this survey of physicians is timely and can provide a baseline for physicians who wish to understand how this new possibility is thought of by other physicians.

This survey can also help to inform the views of others who interact with individuals who want to be preserved. This might include relatives, courts, solicitors, ethicists, journalists, bloggers and others.

The article was well written, neutral in tone, and sought to include all perspectives. As this subject can sometimes elicit significant emotional bias, a treatment that seeks to provide a common basis of facts available to all participants is most welcome.

Reviewer #2: The paper "Physician estimates of the feasibility of preserving the dying for future revival" reports on a robust and comprehensive survey of doctors in the US view on the likelihood of neural information preservation post-cardiac death, their views on their feelings towards post-'death' preservation, and practical steps that they would consider taking to facilitate this post-death revival. The study had particular strengths in soliciting views from a wide range of specialities, and limiting numbers from each speciality in order to prevent skewed-results. The study's design is also strengthened by their exclusion techniques, and the data well interpreted, presented, and potential reasons and correlations given. I would recommend this paper for publication, and suggest two minor changes.

The first paragraph of the discussion is quite long and mainly a repeat of what has come before. I would remove/significantly shorten this. I would also recommend removing the word "cryopreservation" from line 55, and just say "low temperatures are used". Cryopreservation is technically only applicable if preservation is proven.

Reviewer #3: Zeleznikow-Johnston et al. present a cross-sectional survey to evaluate medical perspectives on structural brain preservation and cryonics.

Summary: The survey was rigorously conducted and analyzed, consisting of 13 closed-ended items (11 multiple choice and 2 rating scale combined with open-ended follow-up items in 12 items. 598 validated US physicians responded to the survey invitation, of whom 334 completed the survey and met all inclusion criteria. Of these, 93 physicians rated the possibility of future revival through preservation somewhat or very plausible, 157 as somewhat or very implausible, while 84 were neutral/uncertain. Original survey items and results are available online.

Assessment: This survey is an important and original research contribution, since preservation practices have evolved substantially but largely independently from medical practitioners in the four, respectively six decades since their conceptualization, see [1, 2] for SBP, and [3] for cryonics. Thus far, the academic discussion has taken mainly place in non-clinical disciplines like philosophical bioethics and medical law, see references [1-24], with [25] as a notable exception.

Given the a) medical nature of preservation practices but long evolution independent of medical practitioners, b) psychological obstacles of discussing preservation practices [18] and c) survey results on acceptance of preservation practice in the general population [26, 27] and neuroscientists [28], the present survey fills a long-standing knowledge gap regarding the attitudes of physicians on preservation practices. Medical practitioners are the most proximate sources for the evaluation and provision of such practices for most patients.

The survey data indicate polarized attitudes among physicians with more respondents being either optimistic or pessimistic than neutral. A polarization and emotionalization is particularly evident when looking at open-ended responses, e.g.:

Responseid 9, 60-69 year old general practitioner:

“I wouldn't choose it but, heck, if someone wants to try to cheat death, let them at it. God's plan will prevail eventually. The karmic implications of attempting something like this is an issue that individual will deal with eventually. As previously predicted and commented on, this cannot be offered on someone else's dime. It is NOT a right! To pretend to make it a "right", that would be available to any and all potential persons, then you are proscripting others to provide that service. THIS is immoral and unethical!”

Responseid 62, 30-39 year old psychiatrist:

“If this becomes a realistic outcome in the future, gating it behind wealth would be an extreme delineation from reasonable ethics.”

Responseid 68, 30-39 year old neurosurgeon:

“If access to this is not equal, could become a eugenics nightmare.”

Zeleznikow-Johnston et al. provide a valuable research contribution in documenting this polarization and emotionalization of medical professionals concerning structural brain preservation and cryonics. The controversy around these important but rarely discussed topics requires ongoing discussion and engagement of members of the broader medical community. Therefore, I recommend publication, provided the points below are addressed.

Major comments and suggestions:

Please contextualize the survey with the larger bioethical [1-22], legal [22, 24], psychological [18] and related [29, 30] discussion of structural brain preservation and cryonics.

Directly provide survey population profile and response data in the manuscript in tables as in [26]

Describe how the survey items were designed.

At least briefly analyze or summarize open ended responses in the manuscript.

Analyze gender, age and geography (e.g. as a map).

Discuss acquiescence and question ordering effects.

Line 39-41: “The majority support for pre-mortem anticoagulation and substantial support for pre-cardiac arrest initiation indicate that many physicians would consider accommodating patient requests for preservation-enhancing interventions.”: Please tone down this statement, as the data only supports attitudes towards allowability of anticoagulation.

Line 313 to 322: Uncertainty in CPR is of an aleatoric character while uncertainty in brain preservation and cryonics is of an additional epistemic character. This difference should be acknowledged in the texts when comparing the two.

Line 384-386: "The majority of physicians surveyed support … initiating procedures prior to cardiac arrest." The data indicates only 44.3% supported initiating procedures prior to cardiac arrest.

Line 387-390: "...we recommend proactive development of clinical guidelines and legal frameworks to ensure that patients seeking preservation can receive interventions..." Please tone down advocacy in your conclusion and stay more descriptive.

References:

1. Drexler, E., Engines of Creation. 1986: Fourth Estate.

2. Olson, C.B., A possible cure for death. Med. Hypotheses, 1988. 26(1): p. 77-84.

3. Ettinger, R.C.W., The Prospect of Immortality. 1962.

4. Labaree, L.W. and W.J. Bells, Mr. Franklin: a selection from his personal letters. 1956, New Haven: Yale University Press.

5. Minerva, F., The Ethics of Cryonics: Is It Immoral to Be Immortal? 2018: Palgrave Pivot.

6. Shaw, D., Cryoethics: Seeking life after death. Bioethics, 2009. 23(9): p. 515-521.

7. Moen, O.M., The case for cryonics. J Med Ethics, 2015. 41(8): p. 677-81.

8. Thau, T., Cryonics for all? Bioethics, 2020. 34(7): p. 638-644.

9. Minerva, F. and A. Sandberg, Euthanasia and cryothanasia. Bioethics, 2017. 31(7): p. 526-533.

10. Merkle, R.C., The technical feasibility of cryonics. Med Hypotheses, 1992. 39(1): p. 6-16.

11. Fuhr, G., et al., Unterbrochenes Leben? 2013, Stuttgart, Germany: Fraunhofer Verlag. 142.

12. Cerullo, M.A., The Ethics of Exponential Life Extension through Brain Preservation. Journal of Evolution and Technology, 2016. 26(1): p. 94-105.

13. Cerullo, M.A., Uploading and Branching Identity. Minds and Machines, 2015. 25(1): p. 17-36.

14. Buben, A., Dying to Live: Transhumanism, Cryonics, and Euthanasia, in New Directions in the Ethics of Assisted Suicide and Euthanasia, M. Cholbi and J. Varelius, Editors. 2023, Springer. p. 299-313.

15. Andrade, G. and M.C. Redondo, Cryonics, euthanasia, and the doctrine of double effect. Philosophy, Ethics, and Humanities in Medicine, 2023. 18(1): p. 7.

16. Hayworth, K. Killed by Bad Philosophy. 2010 20 March 2024]; Available from: https://www.brainpreservation.org/content-2/killed-bad-philosophy/.

17. Freitas, R.A. and G.M. Fahy, Cryostasis Revival: The Recovery of Cryonics Patients Through Nanomedicine. 2022: Alcor Life Extension Foundation.

18. German, A. and M. Tretter, Brain Preservation and Cryonics Through the Lens of Moral Psychology. Neuroethics, 2025. 18(1): p. 12.

19. McKenzie, A.T., et al., Structural brain preservation: a potential bridge to future medical technologies. Front Med Technol, 2024. 6: p. 1400615.

20. Shao, J., Cryonics Wager: An Analysis. International Journal of Philosophical Studies, 2025. 33(1): p. 36-49.

21. Stodolsky, D.S., The growth and decline of cryonics. Cogent Soc. Sci., 2016. 2(1): p. 1167576.

22. Mullock, A. and E.C. Romanis, Cryopreservation and current legal problems: seeking and selling immortality. J Law Biosci, 2023. 10(2): p. lsad028.

23. Huxtable, R., Cryonics in the Courtroom: Which Interests? Whose Interests? Medical Law Review, 2018. 26(3): p. 476-499.

24. Spector, D.R., Legal Implications of Cryonics. Cleveland-Marshall Law Review, 1969. 18: p. 341.

25. Whetstine, L., et al., Pro/con ethics debate: when is dead really dead? Crit Care, 2005. 9(6): p. 538-42.

26. Kaiser, S., et al., ATTITUDES AND ACCEPTANCE TOWARD THE TECHNOLOGY OF CRYONICS IN GERMANY. Int J Technol Assess Health Care, 2014: p. 1-7.

27. Rodrigues dos Santos, J.P., et al., Swiss public attitudes to human cryopreservation. medRxiv, 2026: p. 2026.02.16.26346390.

28. Zeleznikow-Johnston, A., E.F. Kendziorra, and A.T. McKenzie, What are memories made of? A survey of neuroscientists on the structural basis of long-term memory. PLOS ONE, 2025. 20(6): p. e0326920.

29. Laakasuo, M., et al., What makes people approve or condemn mind upload technology? Untangling the effects of sexual disgust, purity and science fiction familiarity. Palgrave Communications, 2018. 4(1): p. 84.

30. Laakasuo, M., et al., The dark path to eternal life: Machiavellianism predicts approval of mind upload technology. Personality and Individual Differences, 2021. 177: p. 110731.

.

Reviewer #1: No

Reviewer #2: No

Reviewer #3: No

---

## [Author Response · Author response to Decision Letter 1]

23 Mar 2026

Reviewer #1:

This article addresses questions of growing interest to the medical community. As various forms of preservation become more common, physicians and other members of the medical community will increasingly need to respond appropriately to patients who have a personal interest in the subject. As a consequence, this survey of physicians is timely and can provide a baseline for physicians who wish to understand how this new possibility is thought of by other physicians.

We thank the reviewer for their kind words about our study.

This survey can also help to inform the views of others who interact with individuals who want to be preserved. This might include relatives, courts, solicitors, ethicists, journalists, bloggers and others.

The article was well written, neutral in tone, and sought to include all perspectives. As this subject can sometimes elicit significant emotional bias, a treatment that seeks to provide a common basis of facts available to all participants is most welcome.

We appreciate the reviewer’s kind remarks and hope the study will be useful in this regard.

Reviewer #2:

The paper "Physician estimates of the feasibility of preserving the dying for future revival" reports on a robust and comprehensive survey of doctors in the US view on the likelihood of neural information preservation post-cardiac death, their views on their feelings towards post-'death' preservation, and practical steps that they would consider taking to facilitate this post-death revival. The study had particular strengths in soliciting views from a wide range of specialities, and limiting numbers from each speciality in order to prevent skewed-results. The study's design is also strengthened by their exclusion techniques, and the data well interpreted, presented, and potential reasons and correlations given. I would recommend this paper for publication, and suggest two minor changes.

We thank the reviewer for their kind remarks and for flagging the strengths of our work.

The first paragraph of the discussion is quite long and mainly a repeat of what has come before. I would remove/significantly shorten this.

As per this suggestion, we have now substantially shortened this paragraph.

I would also recommend removing the word "cryopreservation" from line 55, and just say "low temperatures are used". Cryopreservation is technically only applicable if preservation is proven.

We have changed this to “ low temperatures combined with cryoprotective agents with the goal to cryopreserve tissue and minimize ice crystal formation”

Reviewer #3:

Zeleznikow-Johnston et al. present a cross-sectional survey to evaluate medical perspectives on structural brain preservation and cryonics.

Summary: The survey was rigorously conducted and analyzed, consisting of 13 closed-ended items (11 multiple choice and 2 rating scale combined with open-ended follow-up items in 12 items. 598 validated US physicians responded to the survey invitation, of whom 334 completed the survey and met all inclusion criteria. Of these, 93 physicians rated the possibility of future revival through preservation somewhat or very plausible, 157 as somewhat or very implausible, while 84 were neutral/uncertain. Original survey items and results are available online.

This is a fair characterisation of our study.

Assessment: This survey is an important and original research contribution, since preservation practices have evolved substantially but largely independently from medical practitioners in the four, respectively six decades since their conceptualization, see [1, 2] for SBP, and [3] for cryonics. Thus far, the academic discussion has taken mainly place in non-clinical disciplines like philosophical bioethics and medical law, see references [1-24], with [25] as a notable exception.

Given the a) medical nature of preservation practices but long evolution independent of medical practitioners, b) psychological obstacles of discussing preservation practices [18] and c) survey results on acceptance of preservation practice in the general population [26, 27] and neuroscientists [28], the present survey fills a long-standing knowledge gap regarding the attitudes of physicians on preservation practices. Medical practitioners are the most proximate sources for the evaluation and provision of such practices for most patients.

We thank the reviewer for their kind words and agree that our study fills a knowledge gap previously unexplored.

The survey data indicate polarized attitudes among physicians with more respondents being either optimistic or pessimistic than neutral. A polarization and emotionalization is particularly evident when looking at open-ended responses, e.g.:

Responseid 9, 60-69 year old general practitioner:

“I wouldn't choose it but, heck, if someone wants to try to cheat death, let them at it. God's plan will prevail eventually. The karmic implications of attempting something like this is an issue that individual will deal with eventually. As previously predicted and commented on, this cannot be offered on someone else's dime. It is NOT a right! To pretend to make it a "right", that would be available to any and all potential persons, then you are proscripting others to provide that service. THIS is immoral and unethical!”

Responseid 62, 30-39 year old psychiatrist:

“If this becomes a realistic outcome in the future, gating it behind wealth would be an extreme delineation from reasonable ethics.”

Responseid 68, 30-39 year old neurosurgeon:

“If access to this is not equal, could become a eugenics nightmare.”

Zeleznikow-Johnston et al. provide a valuable research contribution in documenting this polarization and emotionalization of medical professionals concerning structural brain preservation and cryonics. The controversy around these important but rarely discussed topics requires ongoing discussion and engagement of members of the broader medical community. Therefore, I recommend publication, provided the points below are addressed.

Major comments and suggestions:

Please contextualize the survey with the larger bioethical [1-22], legal [22, 24], psychological [18] and related [29, 30] discussion of structural brain preservation and cryonics.

We have added the following to the discussion to address this:

“The substantial degree of variation in familiarity with preservation procedures (see Supplementary Table 1) suggests that many physicians are unaware of the broader context in which this form of potential life extension has been proposed and criticised. The basic idea of preserving the bodies of individuals for potential future revival was described by Benjamin Franklin as early as 1773, yet the modern movement is generally traced to a book published in 1962 by Robert Ettinger (16,17), which has stimulated further elaborations in the decades since (2,18–20). These proposals have also been considered in bioethics discussions, both in isolation and in relation to other contentious issues such as euthanasia (21–27). These discussions have not been entirely theoretical, with questions of legal autonomy as it relates to preservation having already been tested in courtrooms(28). Separately from questions of morality and legality, philosophers have also debated which revival methods (if any) would actually constitute a form of survival for the preserved individual, with these discussions situated in broader questions on the nature of personal identity (29–32). There has also been a small amount of social science research into the moral intuitions and personality traits that may drive members of the general public to find preservation more or less desirable and/or socially acceptable (33–36). Notably, the themes that emerged in participants’ open-ended responses - including tensions around patient autonomy and equitable access, debates over whether preservation offers false hope or compassionate hope, and questions about whether revival would constitute genuine survival - closely mirror the central tensions in this broader scholarly discourse.”

Directly provide survey population profile and response data in the manuscript in tables as in [26]

We have now included this as Supplementary Table 1.

Describe how the survey items were designed.

We have added the following text to the Methods:

“Survey items were developed collaboratively by the study authors to address three domains: (1) the perceived feasibility of preservation procedures, (2) clinical interventions that could improve preservation outcomes, and (3) the ethical and legal standing of preservation as an end-of-life option. The final set of items was selected to balance comprehensive coverage of these domains against survey length constraints. Prior to deployment, the survey underwent informal face validity testing with five physician colleagues. Based on their feedback, questions were iteratively revised to improve clarity and comprehension.”

At least briefly analyze or summarize open ended responses in the manuscript.

We have added this as an additional analysis.

In the methods:

“Open-ended responses were extracted from the optional free-text comment fields and filtered to exclude blanks, “None” entries, and responses fewer than 10 characters. The complete response corpus was reviewed in full by a large language model (Claude Sonnet 4.6, Anthropic, March 2026), which identified thematic categories and cross-cutting patterns. To enable reproducible per-comment coding, the language-model-identified themes were operationalised as keyword-based regular expression rules across eight categories (scientific skepticism, patient autonomy, religious/spiritual/framing, equity and access concerns, ethical/moral concerns, legal/regulatory concerns, supportive/optimistic, and death acceptance framing), with a single response able to receive multiple labels. Additional emergent themes not amenable to simple keyword rules were operationalised as more complex pattern sets and are documented in the supplementary coded dataset (open_ended_llm_coded.csv). All qualitative findings are anchored to specific respondent IDs and question labels in that file to permit independent verification.”

In the results:

“Additional commentary through open-ended responses

Of 334 respondents, 221 (66.2%) provided at least one open-ended comment, contributing 1,487 comment instances across 12 questions (mean 6.7 per commenting respondent). The most frequently coded keyword themes were supportive/optimistic (9.8% of responses), patient autonomy (11.4%), legal/regulatory concerns (10.6%), ethical/moral concerns (10.0%), and equity and access concerns (6.9%). Full-corpus qualitative review identified three cross-cutting tensions not well captured by keyword coding. First, an autonomy–equity tension: 22 of 221 commenting respondents (10%) raised both patient-autonomy arguments and wealth-access concerns across their comments, reflecting a conflict internal to the autonomy position itself—“If universally acceptable, preservation should be equally accessible to people regardless of their socioeconomic status; otherwise, this will cause significantly negative impact to human being society and history.” (respondent 188, Q25). Second, a false-hope vs. compassionate-hope debate (present in 19 comments, 1.3%), most prominent in responses to Q20. Third, a consciousness and personal identity problem (present in 23 comments, 1.5%), who questioned whether revival would restore a continuous personal identity regardless of structural preservation quality—“We don’t understand how consciousness works. It’s very difficult to restore something we don’t understand” (respondent 303, Q4).”

Additionally, we altered the section in the limitations discussing our (lack of) analysis of the free comments to read:

“Third, our thematic analysis of qualitative comments relied on large language model assisted coding rather than traditional qualitative methods such as grounded theory or thematic analysis by independent human raters, and the emergent themes identified should be interpreted accordingly.”

Analyze gender, age and geography (e.g. as a map).

We note that gender and age were already subject to some analysis in the original manuscript, as assessed by the ‘Correlations between participant responses’ displayed in Figure 4. However, we did not describe this explicitly in the results. We have added the following to make this clearer:

Physician gender and age were also associated with some attitudinal differences. Male physicians were more familiar with preservation procedures (⍴ = 0.21, p < 10-3), more personally comfortable with patients requesting them (⍴ = 0.17, p = 0.006), more frequently engaged in end-of-life discussions (⍴ = 0.19, p = 0.002), and more supportive of heparin administration prior to cardiac arrest (⍴ = 0.13, p = 0.045). Older physicians similarly reported greater familiarity with preservation procedures (⍴ = 0.23, p < 10-3) and greater support for heparin administration (⍴ = 0.13, p = 0.038). No other significant age or gender based attitudes were observed.

We have also now provided participant responses split based on gender (Supplementary Table 2) and age (Supplementary Table 3).

Regarding geographic effects, we have added Supplementary Figure 1, which displays participant responses by US State. The response pattern reflects the US population distribution, with the vast majority of responses coming from California, New York, Texas and Florida. We note that we did not make a particular effort in this study to perform regional analysis, and that given we lack a large number of responses from each US state, we are underpowered to perform geographic analyses that are more than purely descriptive.

Discuss acquiescence and question ordering effects.

We have added the following to the Limitations section in the discussion:

“Fourth, as we did not run multiple variants of the survey with randomised question presentation order or response order, we cannot entirely rule out acquiescence or question ordering effects. However, we believe the relatively low probability estimate of 25.5% revival success, as well as the substantial opposition described on several items, indicates that acquiescence bias did not dominate participant responses.”

Line 39-41: “The majority support for pre-mortem anticoagulation and substantial support for pre-cardiac arrest initiation indicate that many physicians would consider accommodating patient requests for preservation-enhancing interventions.”: Please tone down this statement, as the data only supports attitudes towards allowability of anticoagulation.

We have changed the sentence to be less ambiguous, writing: “The majority support for pre-mortem anticoagulation and plurality support for pre-cardiac arrest initiation indicate that many physicians would consider accommodating patient requests for preservation-enhancing interventions”. This makes clearer that while support for pre-cardiac arrest was the most common position, it was not a majority.

Line 313 to 322: Uncertainty in CPR is of an aleatoric character while uncertainty in brain preservation and cryonics is of an additional epistemic character. This difference should be acknowledged in the texts when comparing the two.

We have appended the following to the discussion on these lines:

“...(though note that the epistemic uncertainty of preservation’s success is admittedly different to the purely aleatoric uncertainty in cardiopulmonary resuscitation outcomes).”

Line 384-386: "The majority of physicians surveyed support … initiating procedures prior to cardiac arrest." The data indicates only 44.3% supported initiating procedures prior to cardiac arrest.

As per above, we have added the term ‘plurality’ to make this less ambiguous:

“The majority of physicians surveyed support both prescribing medications to optimize preservation outcomes and, in appropriate circumstances, a plurality support initiating procedures prior to cardiac arrest.”

Line 387-390: "...we recommend proactive development of clinical guidelines and legal frameworks to ensure that patients seeking preservation can receive interventions..." Please tone down advocacy in your conclusion and stay more descriptive.

We have weakened this conclusion to state “Given this physic

---

## [Editor Report · Decision Letter 1]

5 Apr 2026

Dear Dr. Zeleznikow-Johnston,

Thank you for submitting your manuscript to PLOS ONE and addressing the reviewer comments. After careful consideration, we feel that it has merit but requires minor edits to fully meet PLOS ONE’s publication criteria as it currently stands. Therefore, we invite you to submit a revised version of the manuscript that addresses the points raised during the review process.

As the reviewers requested a geographical analysis, please incorporate the geographical analysis into the main text by explicitly referring to the new supplementary data and include a brief statement outlining the associated limitations.

As a minor aside, the updated text contains an interesting historical reference to Franklin's comments; however, you may wish to note that Robert Boyle already wrote about the effects of cold to preserve bodies over a century before, and conducted experiments freezing and reviving fish [Boyle (1665) New Experiments and Observations Touching Cold. John Crook, London.]

As the corresponding author, your ORCID iD is verified in the submission system and will appear in the published article. PLOS supports the use of ORCID, and we encourage all coauthors to register for an ORCID iD and use it as well. Please encourage your coauthors to verify their ORCID iD within the submission system before final acceptance, as unverified ORCID iDs will not appear in the published article. *Only* the individual author can complete the verification step; PLOS staff the individual author can complete the verification step; PLOS staff the individual author can complete the verification step; PLOS staff the individual author can complete the verification step; PLOS staff *cannot* verify ORCID iDs on behalf of authors.verify ORCID iDs on behalf of authors.verify ORCID iDs on behalf of authors.verify ORCID iDs on behalf of authors.

We look forward to receiving your revised manuscript.

Kind regards,

Barry L. Bentley, Ph.D.

Academic Editor

PLOS One
---

## [Author Response · Author response to Decision Letter 2]

6 Apr 2026

Editor:

As the reviewers requested a geographical analysis, please incorporate the geographical analysis into the main text by explicitly referring to the new supplementary data and include a brief statement outlining the associated limitations.

We have added the following to the methods, after discussing exclusions based on US state of practice:

“Participant responses by US state are provided in Supplementary Figure 1.”

And we have also added the following limitations material to the discussion:

“Second, we did not make a particular effort to perform regional analysis within the US, and given we lack a large number of responses from each US state, we are underpowered to perform geographic analyses that are more than purely descriptive.”

As a minor aside, the updated text contains an interesting historical reference to Franklin's comments; however, you may wish to note that Robert Boyle already wrote about the effects of cold to preserve bodies over a century before, and conducted experiments freezing and reviving fish [Boyle (1665) New Experiments and Observations Touching Cold. John Crook, London.]

Our thanks for the information about Robert Boyle’s work, as we were previously unaware of this pioneering activity!

We have amended the historical discussion to read: “While experiments on freezing and reviving fish were described by Robert Boyle as early as 1665, and the basic idea of preserving the bodies of humans for potential future revival was described by Benjamin Franklin by 1773, the modern movement is generally traced to a book published in 1962 by Robert Ettinger (16–18), which has stimulated further elaborations in the decades since (2,19–21).”

---

## [Editor Report · Decision Letter 2]

14 Apr 2026

Physician estimates of the feasibility of preserving the dying for future revival

PONE-D-25-67370R2

Dear Dr. Zeleznikow-Johnston,

We’re pleased to inform you that your manuscript has been judged scientifically suitable for publication and will be formally accepted for publication once it meets all outstanding technical requirements.

Kind regards,

Barry L. Bentley, Ph.D.

Academic Editor

PLOS One
---

## [Editor Report · Acceptance letter]

PONE-D-25-67370R2

PLOS One

Dear Dr. Zeleznikow-Johnston,

I'm pleased to inform you that your manuscript has been deemed suitable for publication in PLOS One. Congratulations! Your manuscript is now being handed over to our production team.

Kind regards,

on behalf of

Dr Barry L. Bentley

Academic Editor

PLOS One